# Effects of *Artemisia macrocephala* Jacquem on Memory Deficits and Brain Oxidative Stress in Streptozotocin-Induced Diabetic Mice

**DOI:** 10.3390/molecules27082399

**Published:** 2022-04-08

**Authors:** Atiqul Bari, Syed Muhammad Mukarram Shah, Fakhria A. Al-Joufi, Syed Wadood Ali Shah, Mohammad Shoaib, Ismail Shah, Muhammad Zahoor, Muhammad Naeem Ahmed, Mehreen Ghias, Syed Muhammad Hassan Shah, Atif Ali Khan Khalil

**Affiliations:** 1Department of Pharmacy, University of Swabi, Swabi 23460, Khyber Pakhtunkhwa, Pakistan; atiqulbari@hotmail.com (A.B.); mukaramshah@uoswabi.edu.pk (S.M.M.S.); 2Department of Pharmacology, College of Pharmacy, Jouf University, 72341 Aljouf, Saudi Arabia; faaljoufi@ju.edu.sa; 3Department of Pharmacy, University of Malakand, Dir (Lower), Chakdara 18800, Khyber Pakhtunkhwa, Pakistan; mohammadshoaib13@yahoo.com (M.S.); mehreenghias@yahoo.com (M.G.); 4Department of Pharmacy, Abdul Wali Khan University, Mardan 23200, Khyber Pakhtunkhwa, Pakistan; ismailshah09@gmail.com; 5Department of Biochemistry, University of Malakand, Dir (Lower), Chakdara 18800, Khyber Pakhtunkhwa, Pakistan; 6Department of Chemistry, The University of Azad Jammu & Kashmir, Muzaffarabad 13100, Azad Kashmir, Pakistan; drnaeem@ajku.edu.pk; 7Department of Pharmacy, Sarhad University of Science and Information Technology, Peshawar 25000, Khyber Pakhtunkhwa, Pakistan; syedhassan41@gmail.com; 8Department of Biological Sciences, National University of Medical Sciences, Rawalpindi 46000, Punjab, Pakistan; atif.ali@numspak.edu.pk

**Keywords:** *Artemisia macrocephala* Jacquem, diabetes and memory, cholinergic dysfunction, oxidative stress, discrimination index, biomarkers

## Abstract

Different species of *Artemisia* have been reported to have therapeutic potential in treating various health disorders, including diabetes and memory dysfunction. The present study was planned to evaluate the effects of *Artemisia macrocephala* Jacquem crude extract and its subfractions as antiamnesic agents in streptozotocin-induced (STZ) diabetic mice. The in vivo behavioral studies were performed using the Y Maze test and novel object recognition test (NORT) test at doses of 100 and 200 mg/kg of crude extract and 75 and 150 mg/kg of fractions. The in vitro and ex vivo anticholinesterase activities, along with biochemical parameters (superoxide dismutase, catalase, glutathione and lipid peroxidation) in the brain, were evaluated. Blood glucose levels were monitored with a glucometer; crude extract and fractions reduced the glucose level considerably, with some differences in the extent of their efficacies. The crude extract and fractions demonstrated significant inhibitory activity against cholinesterases (AChE and BuChE) in vitro. Crude, chloroform and ethyl acetate extract were found to be more potent than the other fractions, with IC_50_ of Crd-Am = 116.36 ± 1.48 and 240.52 ± 1.35 µg/mL, Chl-Am = 52.68 ± 1.09 and 57.45 ± 1.39 µg/mL and Et-Am = 75.19 ± 1.02 and 116.58 ± 1.09 µg/mL, respectively. Oxidative stress biomarkers like superoxide dismutase, catalase and glutathione levels were elevated, whereas MDA levels were reduced by crude extract and all fractions with little difference in their respective values. The Y-maze test and novel object recognition test demonstrated declines in memory impairment in groups (*n* = 6) treated with crude extract and fractions as compared to STZ diabetic (amnesic) group. The most active fraction, Chl-Am, was also subjected to isolation of bioactive compounds; three compounds were obtained in pure state and designated as **AB-I**, **AB-II** and **AB-III**. Overall, the results of the study showed that *Artemisia macrocephala* Jacquem enhanced the memory impairment associated with diabetes, elevated acetylcholine levels and ameliorated oxidative stress. Further studies are needed to explore the beneficial role of the secondary metabolites isolated in the present study as memory enhancers. Toxicological aspects of the extracts are also important and need to be evaluated in other animal models.

## 1. Introduction

Nature has been a great source of therapeutic agents in the form of plant secondary metabolites since the beginning of human civilization on earth, and continues to provide mankind with solutions to many health remedies today. In the treatment of Alzheimer’s disease (AD), natural products have been useful in maintaining acetylcholine (ACh) levels, which are pivotal for normal memory function [1,2,3]. A number of epidemiological studies have revealed that diabetes has unfavorable effects on cognitive functions and possibly increases the chances of dementia and related complications in humans. Additionally, the progress of diabetes signifies the conditions of fast-tracked aging. It has been established that diabetes mellitus (DM) is associated with dementia and Alzheimer’s disease [4,5]. Furthermore, memory dysfunction has a strong correlation with oxidative stress [6]. Superoxide dismutase (SOD), catalase (CAT) and glutathione (GSH) are important antioxidant biomarkers, and their levels are lowered in brains under oxidative stress [7]. Similarly, due to peroxidation of lipids in the brain, the level of malondialdehyde (MDA) increases, which is associated with loss of memory [8]. Medicinal plants have been reported to have beneficial effects on brains in conditions of oxidative stress through raising the levels of SOD, GSH and CAT [9] and decreasing the level of MDA [10]. Bioactive compounds from medicinal plants have been investigated for their various biological potentials, as such compounds are compatible with human life, being synthesized from plants as agents of offense and defense. Tremendous research has been undertaken worldwide to explore novel potent compounds and their desired biological potentials, which, apart from affordability, have an excellent safety profile in comparison to synthetic compounds [11]. Traditional systems of medicine are primarily based on plants, and the ingredients in such formulations are not only curative but also nutritive, with fewer side effects [12]. Thus, over the last few decades, physiochemists have been exploring medicinal plants as nootropic agents, capable of treating cognitive dysfunction. A few such plant products are even prescribed by physicians as nootropic agents [11,12].

*Artemisia* is a diverse genus comprising more than 500 species. It belongs to the family Asteraceae. Most species of this genus are found in the temperate zones of North America, Europe and Asia [13,14]. From *Artemisia*, a number of phytochemicals—including terpenoids, flavonoids, glycosides, coumarins, carotenoids, caffeoylquinic acids and sterols—with wide-ranging pharmacological activities have been isolated [15,16]. Several species of this genus have exhibited biological potential, like antihyperlipidemic, antiepileptic, antihypertensive, antidiabetic, anxiolytic, antidepressant, anticancer and hepatoprotective activities. Some species have even been reported to be effective against the COVID-19 virus [17,18,19,20]. Insulin sensitizers, antihyperglycemic and antioxidant agents from plants have been reported to ameliorate the cognitive dysfunction associated with diabetes [21,22]. Aside from the claim, however, no efficient treatment is currently available to manage and/or prevent the cognitive dysfunction associated with DM. There is a need to isolate efficient bioactive compounds with curative effects on DM and its associated cognitive dysfunction.

Therefore, the objective of the current study was to evaluate the therapeutic effects of *Artemisia macrocephala* Jacquem, in the form of extracts, on cognitive and memory impairment, oxidative stress and cholinergic dysfunction in mice. We aimed to evaluate it as antiamnesic agent using both in vivo and ex vivo approaches. The in vitro anticholinesterase activities were also evaluated. To identify responsible compounds in the extracts, silica gel column isolation was applied, resulting in the isolation of three compounds (**AB-I**, **AB-II** and **AB-III**) which will be investigated in our future studies.

## 2. Results

### 2.1. Quantitative Spectrophotometric and Non-Spectrophotometric Phytochemical Analysis

Results of phytochemical analyses of TPC, TFC and TTC (determined spectrophotometrically) of the crude extract (Crd-Am) and fractions of *A. macrocephala* are presented in Table 1. Among them, the chloroform (Chl-Am) and ethyl acetate (Et-Am) fractions exhibited the highest TPC, with mean values of 71.17 ± 0.79 and 63.89 ± 0.83 mg GAE/g of dry sample used, respectively (Table 1). TFC contents were also high in Chl-Am and Et-AM fractions (59.50 ± 1.01 and 51.33 ± 0.98 of mg QE/g, respectively), whereas TTC in mg GAE/g of dry sample were high in Crd-Am (66.31 ± 0.91), The *n*-hexane, butanol and aqueous fractions were found to contain the lowest amounts of phenolics and flavonoids.

Results of alkaloids, flavonoids, saponins and terpenoids, determined through the non-spectrophotometric method, are given in Table 2.

### 2.2. In Vitro Anticholinesterase Activity

The in vitro anticholinesterase inhibitory potential of the crude extract and fractions, along with the IC_50_ values, are given in Appendix A (in the Appendix A).

As shown by the results in Appendix A, it was clear that all extracts demonstrated enzyme inhibitory (AChE and BuChE) activities. The IC_50_ of Crd-Am was found to be 116.36 ± 1.48 and 240.52 ± 1.35 µg/mL; nhex-Am, 934.75 ± 1.21 and 509.26 ± 1.14 µg/mL; Chl-Am, 52.68 ± 1.09 and 57.45 ± 1.39 µg/mL; Et-Am, 75.19 ± 1.02 and 116.58 ± 1.09 µg/mL; But-Am, 254.66 ± 1.34 and 265.33 ± 1.41 µg/mL; and Aq-Am; 502.71 ± 1.28 and 513.66 ± 1.33 µg/mL, respectively, against AChE and BuChE. The standard donepezil demonstrated significant AChE and BuChE inhibition with IC_50_ of 5.04 ± 0.67 µg/mL and 4.01 ± 0.71 µg/mL, respectively.

### 2.3. Acute Toxicity Study

No fatalities (up to an oral dose of 2000 mg/kg b.w. in case of crude extract and 1500 mg/kg b.w.) were observed in acute toxicity studies. As mentioned before, animals were examined each day for two weeks and no signs of convulsion, salivation, diarrhea, sleeping or lethargy were observed. The body weights of all animals in each group were also normal. From the acute toxicity profile and preliminary pharmacological screening, the doses for crude extract selected were 100 and 200 mg/kg b.w., while doses of the fractions of 75 and 150 mg/kg b.w. were subsequently used in the assessment of in vivo pharmacological evaluations. The maximum doses applied were 1/10th of the highest safe dose as described by OECD guidelines.

### 2.4. Y-Maze Spontaneous Alteration

Table 3 shows the results of the Y-maze tests, where STZ significantly lessened the spontaneous alteration from 80.11% to 39.06% (*p* ˂ 0.001, *n* = 6). The spontaneous alteration was normalized by donepezil which increased the %alteration to 79.05 ± 1.60 (*p* ˂ 0.001, *n* = 6) while metformin elevated the %alteration to 74.29 ± 1.96 (*p* ˂ 0.001, *n* = 6). The Crd-Am at doses of 100 and 200 mg/kg b.w. also amplified the %alteration to 57.13 ± 1.59 (*p* ˂ 0.01, *n* = 6) and 62.54 ± 1.50 (*p* ˂ 0.001, *n* = 6), respectively.

Among the fractions, promising effects were produced by Chl-Am at doses of 75 and 150 mg/kg b.w., as compared to amnesic group, found to be 68.57 ± 1.67 (*p* ˂ 0.001, *n* = 6) and 73.89 ± 1.77 (*p* ˂ 0.001, *n* = 6), and Et-Am with 59.37 ± 1.61 (*p* ˂ 0.01, *n* = 6) and 64.69 ± 1.71 (*p* ˂ 0.001, *n* = 6). The above results suggested that Chl-Am was more potent in comparison to other fractions and the crude extract.
molecules-27-02399-t003_Table 3Table 3Effects of crude extract and fractions on spontaneous alternation performance in behavioral Y-maze test.Treatment/Dose (mg)Spontaneous Alternation Performance (%)Normal control80.11 ± 1.65Diabetic control39.06 ± 1.78 ^###^Crd-Am10057.13 ± 1.59 **20062.54 ± 1.50 ***nhex-Am7545.22 ± 1.61 ns15044.78 ± 1.73 nsChl-Am7568.57 ± 1.65 ***15073.89 ± 1.77 ***Et-Am7559.37 ± 1.61 **15064.69 ± 1.71 ***But-Am7547.25 ± 1.56 ns15048.19 ± 1.47 *Aq-Am7546.09 ± 1.40 ns15049.67 ± 1.46 *Donepezil279.05 ± 1.60 ***Metformin5074.29 ± 1.96 ***Mean ± SEM (*n* = 6). One-way ANOVA followed by Dunnett’s post hoc multiple comparison test. ^###^
*p* ˂ 0.001 comparison of STZ-treated diabetic (amnesic) group vs. normal control; * *p* ˂ 0.05, ** *p* ˂ 0.01 and *** *p* ˂ 0.001, ns = non-significant, comparison of STZ-treated diabetic (amnesic) group vs. Donepezil, Metformin, crude extract and fraction-treated groups, using one-way ANOVA followed by Dunnett’s comparison. Crude extract (Crd-Am), hexane (nhex-Am), chloroform (Chl-Am), ethyl acetate (Et-Am), butanol (But-Am) and aqueous (Aq-Am) fractions.

### 2.5. Novel Object Recognition Test

Memory enhancing effects of extracts were evaluated by the novel object recognition test (NORT), results of which are presented in Appendix A (in the Appendix A) and Figure 1. In the sample phase for all tested groups, no changes in exploration time for the objects (in seconds) were observed. During test phase (T2), the exploration time was significantly greater for the novel object (NO) than for the identical object in groups treated with the mentioned doses.

Exploration time for the novel object in seconds was significantly escalated by donepezil (*p* < 0.001, *n* = 6) and lessened for the familiar object with 64.09% index of discrimination (*p* < 0.001, *n* = 6). Crd-Am improved the discrimination index significantly, by 55.81% (*p* < 0.001, *n* = 6) and 57.21% (*p* < 0.001, *n* = 6) at 100 and 200 mg/kg b.w., respectively, when compared to the STZ amnesic group, 32.93%. The discrimination index of chloroform fraction Chl-Am at 75 and 150 mg/kg was increased significantly by 60.86% (*p* < 0.001, *n* = 6) and 62.03% (*p* < 0.001, *n* = 6), respectively.
Figure 1Behavioral results from novel object recognition test (NORT) in short term memory on crude extract and its fractions of *A. macrocephala*. Time spent in sample phase is (**A**) and time in test phase is (**B**), while index of % discrimination, (**C**) is noted for crude extract (200 and 100 mg/kg) and fraction (150 and 75 mg/kg)-treated groups versus STZ-treated diabetic group for the evaluation and recognition of memory in mice models. Mean ± SEM (*n* = 6). One-way ANOVA is followed by Dunnett’s post hoc multiple comparison test. ^###^
*p* ˂ 0.001 comparison of STZ-treated diabetic (amnesic) group vs. normal control, ** *p* ˂ 0.01 and *** *p* ˂ 0.001 as comparison of STZ-treated diabetic (amnesic) group vs. Donepezil-, Metformin-, crude extract- and fraction-treated groups, using one-way ANOVA followed by Dunnett’s comparison. Crude extract (Crd-Am), hexane (nhex-Am), chloroform (Chl-Am), ethyl acetate (Et-Am), butanol (But-Am) and aqueous (Aq-Am) fraction.
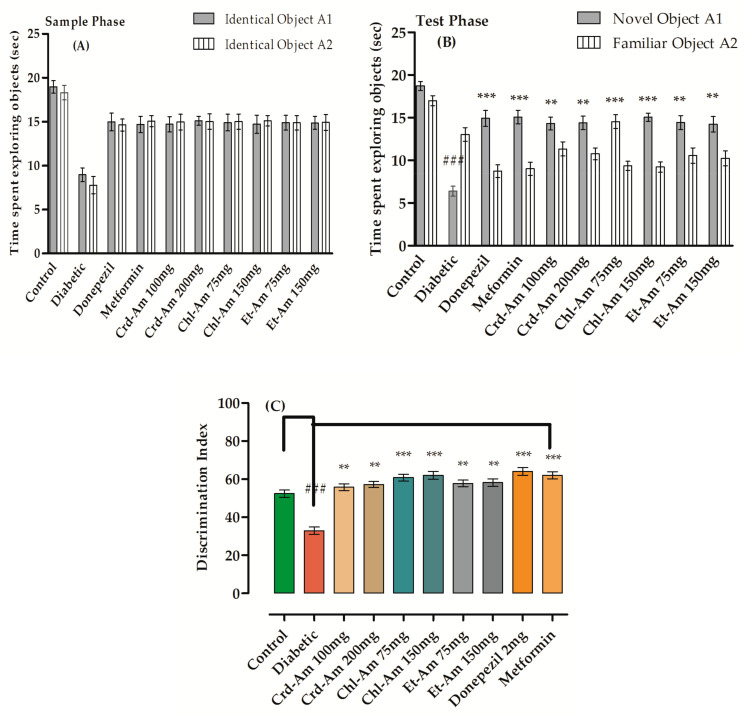


The results of memory enhancement experiments used to evaluate long-term memory using the novel object recognition test (NORT) are given in Appendix A (in the Appendix A) and Figure 2. In the sample phase, for all tested groups, no changes were observed in exploration time (in seconds) for the objects. In the (T2) test phase, the exploration time (in seconds) was considerably greater for the novel object (NO) than for the identical object in the treated groups.

Exploration time for the novel object (in seconds) in donepezil-treated subjects was significantly escalated (*p* < 0.001, *n* = 6), while the same was lessened for the familiar object, with discrimination index of 66.11% (*p* < 0.001, *n* = 6). Crd-Am increased the discrimination index significantly, by 56.72% (*p* < 0.001, *n* = 6) and 58.51% (*p* < 0.001, *n* = 6) at 100 and 200 mg/kg b.w., respectively. This was in comparison to the STZ amnesic group at 32.03%. The chloroform fraction, Chl-Am1, at 75 and 150 mg/kg increased the discrimination index significantly, by 63.39% (*p* < 0.001, *n* = 6) and 64.07% (*p* < 0.001, *n* = 6), respectively, in comparison to the amnesic group at 32.03%.
Figure 2Behavioral results from novel object recognition test (NORT) in long-term memory on crude extract and its fractions of *A. macrocephala*. Time spent in sample phase is (**A**) and time in test phase is (**B**), while index of % discrimination is (**C**) is noted for crude extract (200 and 100 mg/kg) and fraction (150 and 75 mg/kg)-treated groups versus the STZ-treated diabetic group for the evaluation and recognition of memory in mice models. Mean ± SEM (*n* = 6). One-way ANOVA is followed by Dunnett’s post hoc multiple comparison test. ^###^
*p* ˂ 0.001 comparison of STZ-treated diabetic (amnesic) group vs. normal control, ** *p* ˂ 0.01 and *** *p* ˂ 0.001 in comparison with STZ-treated diabetic (amnesic) group vs. Donepezil-, Metformin-, crude extract- and fraction-treated groups, using one-way ANOVA followed by Dunnett’s comparison. Crude extract (Crd-Am), hexane (nhex-Am), chloroform (Chl-Am), ethyl acetate (Et-Am), butanol (But-Am) and aqueous (Aq-Am) fraction.
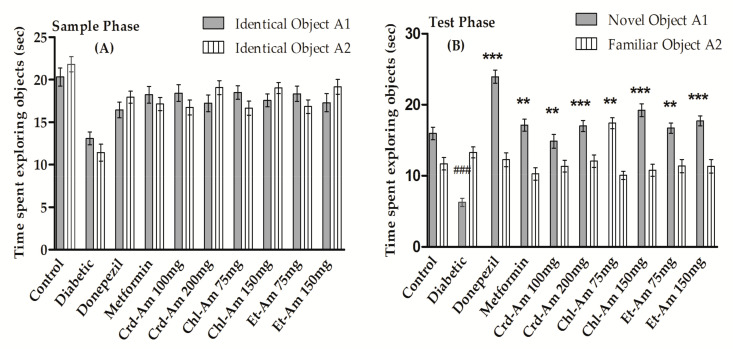

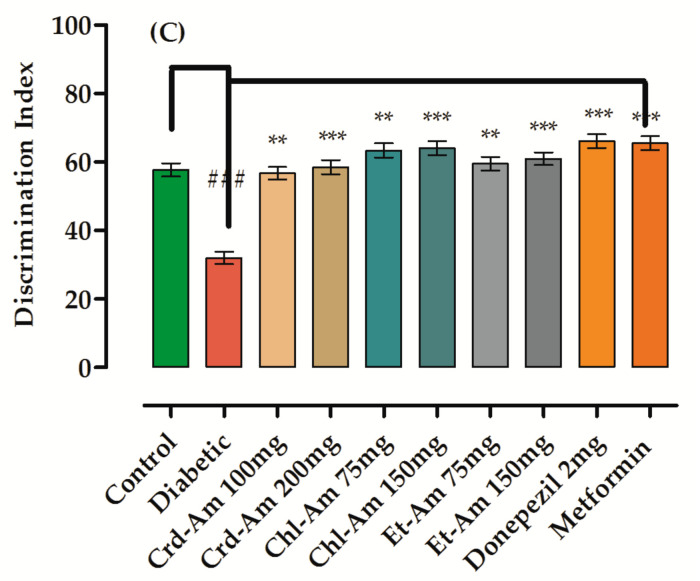


### 2.6. Estimation of Blood Glucose Level, Brain and Body Weight

The results showed that the group treated with STZ had a notable increase in fasting blood glucose (FBG) levels (244.89 ± 2.95, *p* < 0.001, *n* = 6) in comparison to the control group (104.21 ± 2.19, *n* = 6). Administration of crude extract and its fractions significantly (*p* < 0.05, *p* < 0.01 and *p* < 0.001) decreased the FBG level in diabetic mice (Table 4).

### 2.7. Biochemical Parameters and Biomarker Level

After the administration of STZ, substantial elevation in MDA and AChE level and decreased ACh contents were observed. The oxidative stress marker levels were also decreased (CAT and SOD level) in the brain (Appendix A and Table 5). The tested extracts had a distinct declining effect on MDA and AChE, as shown in the mentioned tables. They also enhanced the levels of ACh, CAT and SOD, signifying a possible role as antioxidant.

A considerable increase in the level of AChE in the brain homogenate was observed after STZ administration (Appendix A and Figure 3), which was effectively reversed by the donepezil, metformin, and extracts (*p* < 0.05, *p* < 0.01, *p* < 0.001), signifying the pharmacological function of extracts in treatment of memory impairment, probably via ChE inhibition. Simultaneously, a significant decrease in ACh content was noted in the STZ amnesic group, which was reverted back in the groups treated with extracts and standard.
molecules-27-02399-t005_Table 5Table 5Effects on biomarker levels in brain.SampleTestSOD(U/mg of Protein)CAT(U/mg of Protein)MDAnmol/mg Protein)GSH(μg/mg of Protein)Control9.49 ± 1.3740.18 ± 1.5112.59 ± 1.0656.93 ± 1.71Diabetic3.18 ± 0.93 ^###^11.16 ± 1.37 ^###^38.27 ± 1.44 ^###^29.38 ± 1.66 ^###^Crd-Am100 mg9.01 ± 1.18 **35.81 ± 1.25 **19.11 ± 1.39 ***44.12 ± 1.33 **200 mg9.17 ± 1.21 **36.30 ± 1.16 **18.93 ± 1.22 **45.61 ± 1.41 **Chl-Am75 mg9.39 ± 1.37 ***37.08 ± 1.51 ***17.06 ± 1.43 **47.31 ± 1.49 ***150 mg9.66 ± 1.25 ***38.22 ± 1.17 ***15.81 ± 1.29 ***49.17 ± 1.51 ***Et-Am75 mg9.05 ± 1.03 **35.73 ± 1.39 **18.0 ± 1.21 **45.98 ± 1.47 **150 mg9.21 ± 1.27 ***36.79 ± 1.44 ***17.63 ± 1.11 **47.33 ±1.36 ***Donepezil2 mg10.21 ± 1.41 ***39.08 ± 1.38 ***14.38 ± 1.22 ***51.88 ± 1.34 **Metformin50 mg10.73 ± 1.37 ***40.46 ± 1.33 ***13.89 ± 1.35 ***54.73 ± 1.61 **Mean ± SEM (*n* = 6). One-way ANOVA is followed by Dunnett’s post hoc multiple comparison test. *^###^ p* < 0.001 comparison of STZ-treated diabetic (amnesic) group vs. normal control, ** *p* ˂ 0.01 and *** *p* ˂ 0.001 in comparison with STZ-treated diabetic (amnesic) group vs. Donepezil-, Metformin-, crude extract- and fraction-treated groups using one-way ANOVA followed by Dunnett’s comparison. Crude extract (Crd-Am), chloroform (Chl-Am) and ethyl acetate (Et-Am) fraction.
Figure 3Effects of Crd-Am and fractions on (**A**) AChE and (**B**) ACh levels in brain. Mean ± SEM (*n* = 6). One-way ANOVA is followed by Dunnett’s post hoc multiple comparison test. ^§§§^
*p* < 0.001, ^‡‡‡^
*p* < 0.001 comparison of STZ-treated diabetic (amnesic) group vs. normal control, * *p* ˂ 0.05, ** *p* ˂ 0.01 and *** *p* ˂ 0.001 in comparison with STZ-treated diabetic (amnesic) group vs. Donepezil-, Metformin-, crude extract- and fraction-treated groups using one-way ANOVA followed by Dunnett’s comparison. Crude extract (Crd-Am), chloroform (Chl-Am) and ethyl acetate (Et-Am) fraction.
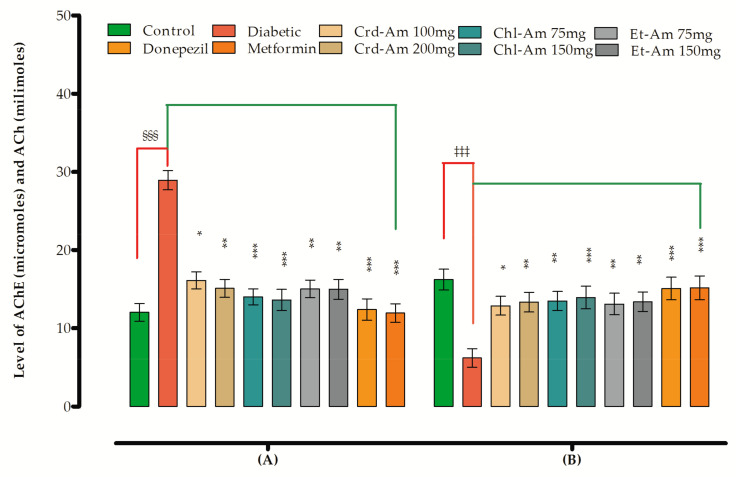


### 2.8. Effects on Biomarker Levels in Brain

The STZ administration resulted in significant declines in levels of SOD by 3.18 ± 0.93 units/mg of protein (about a 2.98-fold increase), *p* < 0.001, *n* = 6 in the brain homogenate, compared to control (9.49 ± 1.37 units/mg protein, *n* = 6), as given in Table 5. The observed decline was overturned by the mice treated with standard donepezil, documented as 10.21 ± 1.71 unit/mg of protein, which was a 3.21-fold increase (*p* < 0.001, *n* = 6) in the brain, as compared to diabetic group of mice. Pretreatment of mice with Crd-Am and fractions significantly increased the SOD level in brain.

Crd-Am significantly increased SOD levels in the brain by 9.01 ± 1.18 51 units/mg protein and 9.17 ± 1.21 units/mg protein at tested doses (2.83 and 2.88-fold increase, respectively; *p* < 0.01, *n* = 6) in comparison to the STZ-treated group. SOD levels in the brain treated with Chl-Am significantly increased to 9.39 ± 1.37 51 units/mg protein and 9.66 ± 1.25 units/mg protein (2.95- and 3.03-fold increase), respectively, at doses of 75 and 150 mg/kg b.w.

As shown in Table 5, it was evident that the Crd-Am and fractions elevated CAT levels in the brains of studied animals with comparable values to that of control group (40.18 ± 1.51 units/mg protein). The STZ administration caused a significant fall of 11.16 ± 1.37 units/mg protein, *p* < 0.001, or a decrease of 3.60-fold, in the levels of catalase enzyme. Donepezil and metformin increased the CAT level significantly, to 39.08 ± 1.38 units/mg protein, *p* < 0.001 (an increase of 3.50-fold) and 40.46 ± 1.33 units/mg protein (an increase of 3.62-fold), respectively, for the tested doses. Crd-Am produced a similar response as the standard (with an increase of 3.20- and 3.25-fold, respectively, for the tested doses). Chl-Am significantly increased the level of catalase, with an increase of 3.32- and 3.42-fold at doses of 75 and 150 mg/kg b.w., respectively. Similar trends were observed when the brain was processed for the determination of GSH level. STZ administration resulted in a significant increase in the level of MDA (an increase of 3.03-fold) that was reverted through administration of donepezil and metformin. Crd-Am decreased its level by 2-fold for each of the tested doses. Chl-Am decreased by 2.24- and 2.42-fold, whereas Et-Am decreased by 2.12- and 2.17-fold, respectively, at tested doses, in comparison to the diabetic amnesic group, where an increase of 3.03-fold was noted.

Streptozotocin significantly affected the level of oxidative stress markers in mice which was demonstrated by decreases in the SOD and CAT levels and increases in the level of MDA. Crd-Am and fractions significantly normalized these marker levels, indicating their possible ameliorating effect on oxidative stress.

### 2.9. Isolated Compounds

The chloroform fraction (being the active fraction, pharmacologically) was subjected to gravity column chromatography using a gradient of increasing polarity for isolation of bioactive compounds in a pure state. Three compounds (**AB-I**, characterized as betasitosterol, **AB-II**, as ursolic acid and **AB-III**, as eupatilin) were obtained in pure states. The chemical structures of the isolated compounds are given in Figure 4.

## 3. Discussion

Acetyl and butyrylcholine are compounds pivotal to nervous transmission in the body. They provide diffuse innervations to the central nervous system [23]. In neurodegenerative diseases, the activities of AChE and BuChE are high, leading to cholinergic deficits [24]. In such conditions, inhibition of AChE and BuChE are critical to attempts to revert the cholinergic deficit. However, apart from a very select few, a 100% efficient inhibitor of the mentioned enzymes has not been found. Plants, as factories of natural products, are under constant exploration all around the world to find these efficient inhibitors. *Artemisia copa* Phil. has been investigated and has shown promising cholinesterases inhibition potential in vitro [25]. The present study was an attempt to confirm the reported results in in vivo and ex vivo experimental animal models. In the preliminary in vitro study, crude (IC_50_ = 116.36 ± 1.48 µg/mL), chloroform (IC_50_ = 52.68 ± 1.09 µg/mL), and ethyl acetate (IC_50_ = 75.19 ± 1.02 µg/mL) fractions of *A. macrocephala* were found to contain efficient inhibitors AChE and BuChE as shown in the observed inhibitory activities in Appendix A.

Preclinical studies have documented and explained learning and memory impairments, including spatial memory and memory retention deficits in rodents and in STZ-induced diabetes in animal models [26]. An association between diabetes and neurodegenerative diseases, such as Alzheimer’s disease, has been well established and associated with a 1.2–1.5-fold greater risk in cognitive function decline [27]. It has been reported that STZ-induced diabetes can produce marked impairment in cognitive function [28,29]. Elevated levels of glucose over a long time play a key role in the onset of behavioral and cognitive complications associated with diabetes. It has been reported that *Artemisia* species have therapeutic potential to be used in the management and treatment of diabetes [30]. Oxidative stress is a leading cause in the development of DM, which then leads to memory problems such as Alzheimer’s disease and related complications [31]. It has also been documented that, in STZ-induced diabetes in animal models, a decline in CAT and SOD level encounters in brain occurs [31]. *A. judaica*, in a reported study, improved the memory deficits and oxidative stress associated with diabetes [32]. In another study, *A. turanica* extract significantly improved oxidative and biochemical marker levels in STZ-induced diabetic rats [32].

From the very beginning of human civilization, extracts and secondary metabolites from plants have been employed in different circumstances as therapeutics. Secondary metabolites of plants are favored as drugs due to their friendliness and broad biological profile. They have proved to be more effective within living systems than synthetic compounds [33]. For centuries, medicinal plants in South Asia have been used as therapeutic agents, especially in areas where most people are poor. The statistics of south Asian countries have shown that medicinal plants are vital in treating diabetes and as neuroprotectants due to their low cost and minimum side effects [34].

Research in the field of phytochemistry has highlighted the occurrence of several commendable compounds, including triterpenes, alkaloids, sterols, polyphenols and flavonoids, that exhibit diversified pharmacological potential such as antioxidant, antidiabetic and anti-Alzheimer’s properties [35]. The extracts tested in this study showed appreciable inhibitory effects on cholinesterases and positive effects on memory impairment.

Administration of wormwood (*Artemisia absinthium* L.) and *Artemisia argyi H* extract helped in restoring enzyme activities involved in the regulation of some neurotransmitters. The extract effectively improved the decline of the SOD level and increased the GSH level in oxidative stress [36,37]. The oxidative stress biomarkers, like SO, CAT and GSH levels in the test animals were elevated by pretreatment with extracts, whereas MDA levels were reduced. Crd-Am increased SOD levels significantly in the brain—by 2.83- and 2.88-fold, respectively at doses of 100 and 200 mg/kg b.w. SOD levels in the brain were significantly increased by Chl-Am, by 2.95- and 3.03-fold, respectively, at doses of 75 and 150 mg/kg b.w. (Table 5).

Crd-Am exhibited an increase in the percentage of spontaneous alternation at 100 and 200 mg/kg as compared to the amnesic group. The latter group was found to be 57.13% and 62.54%, respectively, for the tested doses. Among the fractions (Table 3), promising effects were produced by Chl-Am at doses of 75 and 150 mg/kg, as compared to the amnesic group, which was found to be 68.57% and 73.89%, respectively, followed by Et-Am with 59.37% and 64.69 ± 1.71%, respectively, for the mentioned doses.

In order to find responsible compounds, we isolated three compounds, as shown in in pure form. Although our work is in progress and there are plans to test them in animal models in future studies, currently known facts about the isolated compounds are described herein.

Phytosterols are structurally similar to cholesterol. They are secondary metabolites belonging to the triterpene class of phytochemicals. Beta sitosterol and stigmasterol (important phytosterols) are recognized for their cholesterol-lowering effects as well as their antidiabetic and antioxidant properties. They have also been found to offer boosting effects on the immune system. Beta sitosterol has shown positive effects as a neuroprotector and can enhance cholinergic transmission that is helpful in the restoration of memory [38]. The plant sterols act chemically as antioxidants and physically as stabilizers of membranes—and may also serve as potential candidates in oxidative stress [39]. A number of pharmacological studies have documented that normal use of flavonoids can efficiently enhance cognitive potential in animal models and humans [40]. A number of studies showed that quercetin produced neuroprotection and antagonized oxidative stress when administered to animals [41]. A report by Uriarte-Pueyo and Calvo documented the potentials of 128 flavonoids as cholinesterase inhibitors. Based on their potentials as cholinesterase inhibitors, they were considered to be promising therapeutic agents in the development of new memory enhancers [41,42].

It was proposed from the results that memory impairments are most likely due to elevated acetylcholinesterase activity and deteriorated oxidative status in brain. Overall, our results showed that *A. macrocephala* could be a beneficial therapeutic agent in treating diabetes-associated memory impairment and oxidative stress.

## 4. Materials and Methods

### 4.1. Chemicals

Solvents (methanol, hexane, chloroform, ethyl acetate and butanol) were purchased from Merck (Darmstadt, Germany). AChE Electric eel, BuChE equine serum Lyophilized, 5,5-dithio-bis-nitrobenzoic acid (DTNB), butyryl thiocholine iodide and acetylthiocholine, thiobarbituric acid, Tween-80 and donepezil were procured from Sigma Aldrich, Taufkirchen, Germany, whereas streptozotocin (STZ) powder was obtained from Sigma-Aldrich (St Louis, MO, USA). Metformin was supplied by Medicraft Pharmaceuticals (PVT) Ltd., Hayatabad, Peshawar, Pakistan.

### 4.2. Collection of Plant Material and Authentication

The aerial portion of *A. macrocephala* Jacquem was collected from nearby hills in Chakdara Dir Lower, Khyber-Pakhtunkhwa, Pakistan in April 2020. The plant was authenticated by Professor Dr. Jehandar Shah, Ex-VC UOM and SBBU, and a voucher specimen; BG/AMJ/20-111, was deposited in the herbarium of the mentioned university.

### 4.3. Sample Extraction and Fractionation

Powdered dried plant (7.5 kg) was macerated in methanol at 25 ± 2 °C for two weeks with occasional stirring. The mixture was then filtered. The volume of filtrate was reduced until dry at 40 °C using a rotary evaporator (Rotary Vacuum Evaporator Laborota-4010, Heidolph Co., Schwabach, Germany) to obtain crude extract (Crd-Am, 726 g). The extract was mixed with different solvents to obtain *n*-hexane (nhex- Am, 95 g), chloroform (Chl-Am, 217 g), ethyl acetate (Et-Am, 122 g), butanol (But-Am, 66 g) and aqueous (Aq-Am, 226 g) fractions.

### 4.4. Quantitative Phytochemical Analysis

Crd-Am and subfractions of *A. macrocephala* were evaluated spectrophotometrically and non-spectrophotometrically for major phytochemical groups including total phenolics (TPC), flavonoids (TFC), tannins (TTC), alkaloids, saponins and terpenoids using methods reported in literature [43].

TPC were estimated using Folin–Ciocalteu reagent following standard procedure, whereas spectra were recorded at 765 nm and results were presented as gallic acid equivalents estimated from a calibration curve (Gallic acid equivalents in mg/g of extract). TFC were estimated through a reported procedure involving AICl_3_ as quantifying agent and spectra were recorded at 420 nm. The results were presented as quercetin equivalents in mg/g of extract. TTC were determined through an assay involving reagents (ferric chloride and potassium ferricyanide) using a procedure reported in literature. The absorbance of the reaction mixture was noted at 200 nm and results were expressed as mg of gallic acid equivalents (GAE)/gram [43].

A standardized non-spectrophotometric procedure was applied to confirm the presence of alkaloids, flavonoids, saponins and terpenoids in the extracts. The alkaloids were estimated using 10% acetic acid solution in ethanol followed by addition of conc. ammonium hydroxide solution. For flavonoids, the samples were extracted repeatedly with 80% aqueous methanol, filtered and evaporated according to the reported procedure in literature. The saponins were extracted by successive treatment with di-ethyl ether in a separating funnel, followed by *n*-butanol treatment. The *n*-butanol combined fraction was washed twice with 1 mL of 5% aqueous NaCl. The solution was evaporated in a water bath and percent yield of saponins was estimated. Total terpenoids in the sample were quantified by soaking the sample in alcohol for 24 h, followed by filtration and thorough extraction with petroleum ether. The percent yield of terpenoids was then calculated following reported guidelines [43].

### 4.5. Isolation of Bioactive Compounds

After in vitro and in vivo evaluation, the pharmacologically active fraction, the chloroform (Chl- Am) fraction, was subjected to gravity column chromatography with pure *n*-hexane with ethyl acetate as eluent (mobile phase), resulting in 34 fractions based on TLC analysis. Fractions having same Rf values were combined and again loaded to a narrow pen column to obtain compounds in pure state. Three compounds, designated as **AB-I**, **AB-II** and **AB-III**, were obtained. The compound **AB-I** was characterized as betasitosterol (phytosterol), **AB-II** as ursolic acid (pentacyclic triterpenoid) and **AB-III** as eupatilin (flavonoid). They were authenticated by spectroscopic techniques, including 1H, 13CNMR and mass spectrometry, as well as by comparison with spectral data already published [20,44].

### 4.6. Animals and Ethical Approval

Balb/C male mice (19–24 g) were procured from the National Institute of Health, Islamabad, Pakistan, and housed in plastic cages in an animal house under standard conditions (55–65% relative humidity, 25 ± 2 °C and 12 h light and dark cycle). They were provided with water and standard diet ad libitum. The Departmental Ethical Committee of the University approved all the protocols used in the study (Pharm/EC-Am/16-10/20) that were in agreement with the University of Malakand Animal Bye-Laws 2008, Scientific Procedures Issue-I.

### 4.7. In Vitro Anticholinesterase Activity

To assess the inhibitory effects of acetylcholinesterases by extract, the enzyme AChE, a solution of Ellman’s Reagent (DTNB) and acetylthiocholine iodide (ATChI, substrate) and the extract’s concentrations (62.5–1000 micrograms) [45] were reacted following standard protocol. In similar fashion, butyrylcholinesterase inhibitory potentials were evaluated using enzyme BuChE, DTNB and butyryl thiocholine Iodide (BTChI), and the absorbance at 412 nm was recorded. Donepezil was taken as standard and the obtained data, in triplicate, were used to estimate IC_50_ of each extract [46].

### 4.8. Acute Toxicity Test

The extracts were monitored for possible toxicological effects following standard procedure [28]. Different doses of crude extract (2000 and 5000 mg/kg b.w.) and fractions (1500 and 3000 mg/kg b.w.) were given orally to groups of mice (*n* = 6). As negative control, Tween-80 (2%) in normal saline was administered. Each mouse in the group was carefully examined for mortality or any adverse effects for 24 h, followed by examination for two weeks, during which water and food were provided according to the described procedure. Animals were checked on a daily basis for signs of diarrhea, convulsions, lethargy, sleeping, salivation and tremor for two weeks. Their body weight was also measured on a weekly basis. The crude extract remained safe and nontoxic at a dose range of 2000 mg/kg b.w. while the fraction was found safe at 1500 mg/kg b.w. Selection of doses for in vivo pharmacological assessment of cognitive function using animal model was carried out from in vivo toxicological studies as per OECD (Organization for Economic Cooperation and Development; 2001) Guideline and approach to practical acute toxicity testing by Dietrich Lorke (1983) and Animal Research Reporting *In Vivo* Experiments (ARRIVE) guidelines. With the toxicity data at hand, effective doses (mg/kg b.w.) were selected for behavioral studies after preliminary pharmacological assessment, as per the OECD guidelines. Crd-Am, at doses of 100 and 200 mg/kg b.w. (maximum dose was 1/10th of the safe dose), was given to experimental animals. The subfractions, at doses of 75 and 150 mg/kg b.w., were also given (the maximum dose is 1/10th of the safe dose) [47,48].

### 4.9. Induction and Assessment of Diabetes

Diabetes was induced by streptozotocin (STZ, 50 mg/kg, i.p.) in a citrate buffer to overnight-fasted male mice. After 72 h of administration, blood glucose levels in blood collected from tail vein, were measured using a glucometer (ACCU-CHECK, Active blood glucose meter, Korea). Those animals having fasting levels of blood glucose higher than 250 mg/dL were subjected to assessment of memory [49].

#### 4.9.1. Drug Treatment and Assessment of Memory

A total of 96 Balb/C mice were arbitrarily divided into groups comprising 6 animals each, as follows: Control group A (Con): normal mice treated with Tween-80.

Diabetic Control group B (D Con): STZ induced diabetic mice treated with Tween-80.

Diab+ crude extract group C (Crd-Am): diabetic mice treated with crude extract (Crd-Am) (100 mg/kg).

Diab+ crude extract group D (Crd-Am): diabetic mice treated with crude extract (Crd-Am) (200 mg/kg).

Diab+ hexane group E (nhex-Am): diabetic mice treated with hexane fraction (nhex-Am) (75 mg/kg).

Diab+ hexane group F (nhex-Am): diabetic mice treated with hexane fraction (nhex-Am) (150 mg/kg).

Diab+ chloroform group G (Chl-Am): diabetic mice treated with chloroform fraction (Chl-Am) (75 mg/kg).

Diab+ chloroform group H (Chl-Am): diabetic mice treated with chloroform fraction (Chl-Am) (150 mg/kg).

Diab+ ethylacetae group I (Et-Am): diabetic mice treated with ethyl acetate fraction (Et-Am) (75 mg/kg).

Diab+ ethylacetae group J (Et-Am): diabetic mice treated with ethyl acetate fraction (Et-Am) (150 mg/kg).

Diab+ butanol group K (But-Am): diabetic mice treated with butanol fraction (But-Am) (75 mg/kg).

Diab+ butanol group L (But-Am): diabetic mice treated with butanol fraction (But-Am) (150 mg/kg).

Diab+ aqueous group M (Aq-Am): diabetic mice treated with aqueous fraction (Aq-Am) (75 mg/kg).

Diab+ aqueous group N (Aq-Am): diabetic mice treated with aqueous fraction (Aq-Am) (150 mg/kg).

Donepezil group O (Dnzo): diabetic mice treated with Donepezil (2 mg/kg).

Metformin group P (Mtfm): diabetic mice treated with Metformin (50 mg/kg).

The animals in group A and B served as control group and diabetic control group, respectively. The animals in respective groups C–N were administered with various doses via oral gavage (p.o.), whereas group O and P were positive control (donepezil and metformin) and received treatment via oral route. The doses were given once a day until the conclusion of behavioral studies (after 28 days). All the behavioral tests were performed between 8:00 and 12:00 a.m. to avoid any false results in the performance of the animals. The apparatus used in the behavioral study was cleaned and sanitized before each experiment. The novel object recognition test (NORT) and Y-maze test were performed to evaluate the effect of the extracts on memory impairment associated with DM. For ex vivo evaluation of biochemical markers in the brain, after experimentation, the animals were sacrificed by euthanasia with isoflurane. Figure 5 summarizes all the experimental details.

#### 4.9.2. Y-Maze Test

A single session of Y-Maze was applied to record spontaneous alternations in the behavior of the animals (assessing the exploratory behavior and instant memory functioning). The apparatus used had three arms in Y shapes (35 cm × 8 cm × 15 cm) with an equilateral triangular central area. Each mouse, previously naive to the maze, was placed at the terminal end of one arm. The animals were allowed to move and explore freely for 8 min in the maze and the number of arm entries was noted. One arm entry was considered complete when the hind paws of the animal were completely inside the arm. The spontaneous alternation (as a percentage) was calculated following the described procedure in literature [50].

#### 4.9.3. Novel object Recognition Test (NORT)

The slightly modified apparatus consisted of a box made up of white colored plexiglass (40 cm × 40 cm × 30 cm) with two objects (differing in shape and made of plastic_ placed within to be discriminated by the animals. After the acclimatization period, one day prior to the test, animals were permitted to explore any object in the apparatus for about 2–5 min to habituate. For short term memory evaluation, the test was performed 5 min after sample administration, while for long-term memory, the test was performed 24 h after sample application. On the day of the experiment (sample phase, T1), two objects (identical) were placed in the apparatus at two corners opposite each other and time, taken in seconds spent exploring or recognizing the objects, was noted. The exploration was considered complete if a mouse touched that object or pointed its nose from it at a distance less than 2 cm. The test phase (T2) was conducted after 24 h of the sample phase (T1). A new (novel) object was placed instead of the old one. The animals were kept in the same fashion as in T1 and the exploration time for the new object (N) and the familiar object (F) were noted separately. The relation, (N − F/ N + F) × 100, was used to determine discrimination index (DI) [51].

#### 4.9.4. Estimation of Blood Glucose Level, Brain and Body Weight

After 24 h of behavioral study, fasting blood glucose levels of each group were determined using a glucometer (ACCU-CHECK, Active Blood Glucose Meter). Blood samples were collected by cutting the tail tip of the mice [52]. At the end of the behavioral analysis, all the animals were guillotined and the skulls were cut open to quickly extract and weigh the whole brain [53].

#### 4.9.5. Biochemical Estimation

The brain was extracted and a homogenate (10% *w*/*v*) was made with phosphate buffer (0.1 M, pH 7.4), followed by centrifugation to obtain a supernatant for the assessment of levels/activity of AChE, ACh, CAT, GSH, SOD and MDA [54].

#### 4.9.6. Statistical Analysis

The data in this manuscript are expressed as mean ± SEM and have been statistically analyzed with software (GraphPad Prism 5.01 version) using analysis of variance (ANOVA) and Dunnett’s test. *p* < 0 05 was considered statistically significant.

## 5. Conclusions

The crude extract and fractions of *A. macrocephala* possessed high contents of TPC, TFC and TTC. All other major phytochemical groups were also found in the extracts. The in vivo behavioral and ex vivo oxidative stress biomarkers in the brains of mice were evaluated. The in vitro anticholinesterase potentials of the extracts were also evaluated, and it was noted that potential inhibitors of AChE and BChE could be present in the extracts. Treatment with *A. macrocephala* ameliorated memory impairment in behavioral tests and provided significant protection from oxidative stress in the brain. The current study concluded that *A. macrocephala* could be an encouraging plant resource and would be a potent neuropharmacological drug candidate against neurodegenerative diseases associated with diabetes. Moreover, the isolated compounds are under further investigation to underpin their neuroprotective potential, associated with diabetes.

## Figures and Tables

**Figure 4 molecules-27-02399-f004:**
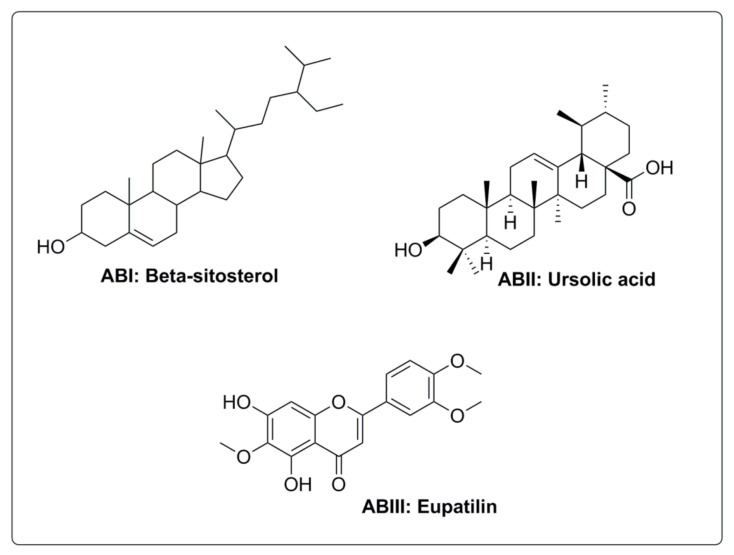
Isolated compounds from the pharmacologically-active fraction, Chl-Am.

**Figure 5 molecules-27-02399-f005:**
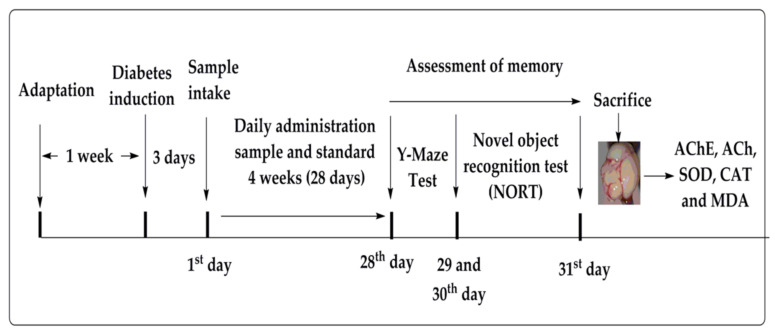
Experimental procedure employed for the assessment of memory in STZ-induced diabetic mice.

**Table 1 molecules-27-02399-t001:** Spectrophotometric quantitative phytochemical analysis of crude extract and fractions.

Sample	TPC (mg GAE/g)	TFC (mg QE/g)	TTC (mg GAE/g)
Crd-Am	59.11 ± 0.53	49.31 ± 0.56	66.31 ± 0.91
Nhex-Am	36.09 ± 0.73	28.21 ± 0.71	14.09 ± 0.73
Chl-Am	71.17 ± 0.79	59.50 ± 1.01	39.70 ± 0.88
Et-Am	63.89 ± 0.83	51.33 ± 0.98	38.21 ± 0.67
But-Am	43.13 ± 0.91	33.26 ± 0.67	48.67 ± 1.01
Aq-Am	34.27 ± 0.67	28.19 ± 0.89	42.15 ± 0.71

All values are expressed as mean ± SEM, *n* = 3, TPC: total phenolic contents; TFC: total flavonoid contents; TTC: total tannin contents; Crd-Am: crude extract; nhex-Am: *n*-hexane fraction; Chl-Am: chloroform fraction; Et-AM: ethyl acetate fraction; But-Am: butanol fraction; Aq-Am: aqueous fraction.

**Table 2 molecules-27-02399-t002:** Non-spectrophotometric quantitative phytochemical analysis of crude extract and fractions.

Sample	Yield (%)			
	Alkaloids	Flavonoids	Saponins	Terpenoids
Crd-Am	4.11 ± 0.48	8.63 ± 0.86	4.75 ± 0.73	8.03 ± 0.79
nhex-Am	0.88 ± 0.33	2.07 ± 0.61	3.72 ± 0.65	6.54 ± 0.81
Chl-Am	3.09 ± 0.28	8.58 ± 0.87	4.22 ± 0.73	7.39 ± 0.79
Et-Am	2.11 ± 0.39	8.01 ± 0.73	3.97 ± 0.67	7.05 ± 0.71
But-Am	1.13 ± 0.71	4.11 ± 0.67	2.90 ± 0.57	4.17 ± 0.58
Aq-Am	1.68 ± 0.67	2.39 ± 0.61	2.04 ± 0.38	5.96 ± 0.71

All values are expressed as mean ± SEM, *n* = 3.

**Table 4 molecules-27-02399-t004:** Effects of crude extract and fractions of *A. macrocephala* on blood glucose level, brain and body weight of mice.

Sample Test	Dose (mg/kg b.w.)	Body Weight	Brain Weight	Blood Glucose Level
Control		24.21 ± 1.12	9.09 ± 2.19	104.21 ± 2.19
Diabetic		20.39 ± 2.01	12.11 ± 1.98 ^†††^	244.89 ± 2.95 ^†††^
Crd-Am	100	23.19 ± 1.76	9.79 ± 1.03 ***	143.94 ± 2.19 *
	200	24.11 ± 1.14	10.51 ± 1.91 ***	131.18 ± 2.13 **
Chl-Am	75	23.39 ± 1.06	9.96 ± 1.41 *	135.76 ± 2.22 *
	150	23.14 ± 1.11	10.19 ± 1.21 **	130.92 ± 1.84 **
Et-Am	75	23.29 ± 1.15	9.89 ± 1.15 *	140.86 ± 1.67 **
	150	22.59 ± 1.12	10.11± 1.97 **	135.01 ± 1.91 **
Donepezil	2	24.18 ± 1.17	10.13 ± 1.15 ***	196.88 ± 2.38 ***
Metformin	50	24.89 ± 1.91	9.91 ± 1.51 ***	162.01 ± 2.83 ***

Mean ± SEM (*n* = 6). One-way ANOVA is followed by Dunnett’s post hoc multiple comparison test. ^†††^
*p* < 0.001 comparison of STZ-treated diabetic (amnesic) group vs. normal control, * *p* ˂ 0.05, ** *p* ˂ 0.01 and *** *p* ˂ 0.001 in comparison with STZ-treated diabetic (amnesic) group vs. Donepezil-, Metformin-, crude extract- and fraction-treated groups, using one-way ANOVA followed by Dunnett’s comparison. Crude extract (Crd-Am), chloroform (Chl-Am) and ethyl acetate (Et-Am) fraction.

## Data Availability

Not applicable.

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
