# Peer review of "Effects of Artemisia macrocephala Jacquem on Memory Deficits and Brain Oxidative Stress in Streptozotocin-Induced Diabetic Mice"

_molecules, 2022, doi:10.3390/molecules27082399_

Round 1

Reviewer 1 Report

  1. Whether it is more intuitive and beautiful to process the results in the form of a statistical graph.
  2. It is unclear how the doses of 100 and 200 mg/kg of crude extract were determined in acute toxicity experiments.
  3. How to determine the dose of streptozotocin-induced diabetes in mice is 50mg/kg.
  4. The Figure 2(B)is wrongly marked, and it is marked the same.
  5. The Figure 3(B)is wrongly marked, and it is marked the same.

Author Response

Reviewer 1:

  1. Reviewer question #1:  Whether it is more intuitive and beautiful to process the results in the form of a statistical graph.
  • Author response: Worthy reviewer, thanks for the valuable suggestion, only graphs were retained in the main text where Table 3, Table 5, Table 6 and Table 8 were moved to Supplementary information as Table S1, S2, S3 and S4 respectively. Now only 5 tables are in article file.

2. Reviewer question #2: It is unclear how the doses of 100 and 200 mg/kg of crude extract were determined in acute toxicity experiments.

  • Author response: Worthy reviewer, the statement in acute toxicity has been modified and dose selection has been added instead as per previous preliminary pharmacological data and previous published reports.

Selection of doses for in-vivo pharmacological assessment of cognitive function using animal model was carried out from in-vivo toxicological studies as per OECD (Organization for Economic Cooperation and Development) (2001) Guideline and approach to practical acute toxicity testing by Dietrich Lorke (1983) and Animal Research: Reporting In-Vivo Experiments (ARRIVE) guidelines. With the toxicity data at hand, effective doses (mg/kg b.w) were selected for behavioral studies after preliminary pharmacological assessment in our laboratory as well as published data elsewhere. The findings of preliminary pharmacological activity lend a hand to standardize the Crd-Am extract and fractions for assessment and selection of doses for pharmacological investigation [29,30]. According to the information provided in literature the testing doses may be 1/10th of highest dose tested in toxicity study.

3. Reviewer question #3: How to determine the dose of streptozotocin-induced diabetes in mice is 50 mg/kg.

  • Author response: Worthy reviewer, the dose of streptozotocin-induced diabetes in mice is 50 mg/kg that has been established. Please check following literature where the mentioned dose has been used.
  • Al-Salmi FA, Hamza RZ. Efficacy of Vanadyl Sulfate and Selenium Tetrachloride as Anti-Diabetic Agents against Hyperglycemia and Oxidative Stress Induced by Diabetes Mellitus in Male Rats. Current Issues in Molecular Biology. 2021 Dec 24;44(1):94-104.
  • Nazir N, Zahoor M, Nisar M, Khan I, Karim N, Abdel-Halim H, Ali A. Phytochemical analysis and antidiabetic potential of Elaeagnus umbellata (Thunb.) in streptozotocin-induced diabetic rats: pharmacological and computational approach. BMC complementary and alternative medicine. 2018 Dec;18(1):1-6.
  • Nazir N, Zahoor M, Ullah R, Ezzeldin E, Mostafa GA. Curative effect of catechin isolated from Elaeagnus umbellata Thunb. berries for diabetes and related complications in streptozotocin-induced diabetic rats model. Molecules. 2020 Dec 30;26(1):137.
  • Araya TY, Karim A, Hailu GS, Periasamy G, Kahsay G. Antihyperglycemic Activity of TLC Isolates from the Leaves of Aloe megalacantha Baker in Streptozotocin-Induced Diabetic Mice. Diabetes, Metabolic Syndrome and Obesity: Targets and Therapy. 2021;14:1153.

4. Reviewer question #4: The Figure 2(B)is wrongly marked, and it is marked the same.

  • Author response: Worthy reviewer, thanks for valuable suggestion. The portion has been modified accordingly

5. Reviewer question #5: The Figure 3(B)is wrongly marked, and it is marked the same.

  • Author response: Worthy reviewer, thanks for valuable suggestion. The portion has been modified accordingly

Reviewer 2 Report

The aim of this study was to investigate effects of Artemisia macrocephala Jachem (AMJ) on memory deficits and brain oxidative stress in streptozotocin-induced diabetic mice based on the observation that diabetes mellitus increases the risk of cognitive impairments and development of Alzheimer's disease (AD). The authors prepared crude extract and several extracts from various solvents. They found that these extract, in various degrees, inhibit cholinesterases, improve behavioural response in novel object recognition test and Y maze test, positively affect markers of oxidative stress (superoxide dismutase, catalase, glutathione, malondialdehyde) and blood glucose level, indicating that AMJ could be effective against DM and AD.

Although the study is interesting, there are several important issues that should be resolved before considering acceptance.

At first, English should be revised thoroughly (professional assistance is suggested). There are numerous grammar mistakes throughout the manuscript and examples of sentences that are difficult to follow. I put some examples bellow, but the manuscript needs extensive revision for language and grammar.

Abstract is too long, with too many details. It is not correct that „Superoxide dismutase, catalase levels were creased and glutathione level was reduced by crude extract…..” (line 40)

Line 279 – „Similarly, the results of TTC in mg GAE/g are also shown in Table 1. On the other hand, n-hexane, butanol and aqueous fractions were found to have lowest contents of phenolics, flavonoids and tannins.” – This is NOT correct; as indicated in Table 1, TTC content has the highest values in Crd-Am and But-Am; TTC is NOT similar with TPC and TFC

Table 3 – Results presented in Table 3 are not acceptable in this form; they should be represented as graph(s) (nonlinear regression, curve fit in GraphPad Prism) to be more informative and ensure visual comparison of cholinesterase inhibitory capacity of different extracts.

Although it seems  that there are many data presented in this manuscript, in same cases, the same results are presented in table, figures and in the main text again – this should be avoided; something could be moved to supplementary

  • lines 298-302 and Table 3
  • Table 8, Figure 4
  • Lines 441-445–all the numbers are indicated in the table and the main text

Discussion section should be carefully rewritten. Instead of repeating Introduction, the authors should put more attention on potential mechanisms of action and comparison with similar studies using plants, extracts, or active components from Artemisia species

Lines 525-526 – “The oxidative stress biomarkers like superoxide dismutase, catalase levels were elevated by extracts whereas glutathione level was reduced. “– NOT correct for GSH (Table 9)

Line 515 - …”for this reason are more effective” – please found appropriate references for this statement

Minor issues

Description of x-axis for Figure 4 should be improved (at least changed to horizontal orientation)

The legend for familiar object A2 object in Figure 2 and Figure 3 (test phase) should be corrected

Lines 406-412– tables where the results are presented should be indicated in the main text

3.8 should not be separated from 3.7 in the Results section

Figures 2 and 3, Lines 340 and 372 – fractions should be specified

Line 469 - Scopolamine in this study?

Metformin should be added to 2.1. Chemicals

Lines 201,203,205,207,209, …….. – animals were not treated with hexane, chloroform, ethyl acetate, butanol,…….   ; they were treated with hexane, chloroform, ethyl acetate, butanol fraction

Line 57, 63-64 – repetition

Line 61 – diabetes mellitus (DM)

Line 76 – Artemisia belongs to  Asteraceae

Line 119 – ( instead of I

Line 251 – objects were identical or similar?

Line 348 - in comparison to amnesic group 32.93%. – could be deleted

Line 400 – FBG should be defined

Lines 525-526 – abbreviations should be used

English

Line 70 – „The researchers are trying to isolate novel potent compounds with easy affordability

and safety profile than synthetic drugs for low income patients [11].”

Lines 209, 211, 359, 366, 402, 425, 432, 450 – change ethylacetae to ethyl acetate  

Line 28 – streptozotocin-induced diabetic mice

Line 58 – epidemiological studies have revealed

Line 63 - ACh possesses

Table 4 Line 318 – “Crd-Am exhibited the major rise (PË‚0.01, PË‚0.001) in the spontaneous alternation at 100 and 200 mg/kg as compared to amnesic group that was found to be 57.13±1.59 (PË‚0.01, n=6) and 62.54±1.50 (PË‚0.001, n=6).” – what is meant by the major rise as some other extracts demonstrated higher effects?

Line 124 - AICl3 as quantifying

Line 184 – in a citrate buffer

Line 187 - Those animals having fasting blood glucose level higher than 250 mg/dL were subjected to assessment of memory [29].

Table 2 – yield instead of yield

Line 302 - respectivel against – please correct English

Line 487 – hippocampal

Author Response

Reviewer 2:

The aim of this study was to investigate effects of Artemisia macrocephala Jachem (AMJ) on memory deficits and brain oxidative stress in streptozotocin-induced diabetic mice based on the observation that diabetes mellitus increases the risk of cognitive impairments and development of Alzheimer's disease (AD). The authors prepared crude extract and several extracts from various solvents. They found that these extract, in various degrees, inhibit cholinesterases, improve behavioural response in novel object recognition test and Y maze test, positively affect markers of oxidative stress (superoxide dismutase, catalase, glutathione, malondialdehyde) and blood glucose level, indicating that AMJ could be effective against DM and AD.

Although the study is interesting, there are several important issues that should be resolved before considering acceptance.

At first, English should be revised thoroughly (professional assistance is suggested). There are numerous grammar mistakes throughout the manuscript and examples of sentences that are difficult to follow. I put some examples bellow, but the manuscript needs extensive revision for language and grammar.

  • Thank you worthy reviewer, for the positive input. The paper has been revised with help of language expert and hope it will be ok now.

Reviewer question #1:  Abstract is too long, with too many details. It is not correct that „Superoxide dismutase, catalase levels were creased and glutathione level was reduced by crude extract…..” (line 40)

  • Author response: Worthy reviewer, the abstract has been modified accordingly and the statement has been corrected accordingly as per valuable suggestion.

Reviewer question #2:  Line 279 – „Similarly, the results of TTC in mg GAE/g are also shown in Table 1. On the other hand, n-hexane, butanol and aqueous fractions were found to have lowest contents of phenolics, flavonoids and tannins.” – This is NOT correct; as indicated in Table 1, TTC content has the highest values in Crd-Am and But-Am; TTC is NOT similar with TPC and TFC

  • Author response: Worthy reviewer, thanks for valuable suggestion. The mistakes has been modified accordingly.

Reviewer question #3:  Table 3 – Results presented in Table 3 are not acceptable in this form; they should be represented as graph(s) (nonlinear regression, curve fit in GraphPad Prism) to be more informative and ensure visual comparison of cholinesterase inhibitory capacity of different extracts.

  • Author response: Worthy reviewer, Table 3 has been moved as Supplementary information as Table S1

Reviewer question #4:  Although it seems  that there are many data presented in this manuscript, in same cases, the same results are presented in table, figures and in the main text again – this should be avoided; something could be moved to supplementary

  • lines 298-302 and Table 3
  • Table 8, Figure 4
  • Lines 441-445–all the numbers are indicated in the table and the main text
  • Author response: Worthy reviewer, the Table 3, Table 5, Table 6 and Table 8 has been moved to supplementary file and numbered as Table S1, S2, S3 and S4.

Reviewer question #5:  Discussion section should be carefully rewritten. Instead of repeating Introduction, the authors should put more attention on potential mechanisms of action and comparison with similar studies using plants, extracts, or active components from Artemisia species

  • Author response: Worthy reviewer, the whole section was rephrased accordingly. Comparison with similar studies were accordingly made. Hopefully it will be ok now.

Reviewer question #6:  Lines 525-526 – “The oxidative stress biomarkers like superoxide dismutase, catalase levels were elevated by extracts whereas glutathione level was reduced. “– NOT correct for GSH (Table 9)

  • Author response: Worthy reviewer, thanks for valuable suggestion. Corrections were made accordingly.

Reviewer question #7:  Line 515 - …”for this reason are more effective” – please found appropriate references for this statement

  • Author response: Worthy reviewer, the reference has been added for the statement as per valuable suggestion.

Minor issues

Reviewer question #1:  Description of x-axis for Figure 4 should be improved (at least changed to horizontal orientation)

  • Author response: Worthy reviewer, description of x-axis for Figure 4 has been corrected as per valuable suggestion.

Reviewer question #2:  The legend for familiar object A2 object in Figure 2 and Figure 3 (test phase) should be corrected

  • Author response: Worthy reviewer, thanks for valuable suggestion. The legend for familiar object A2 has been modified accordingly.

Reviewer question #3:  Lines 406-412– tables where the results are presented should be indicated in the main text

  • Author response: Worthy reviewer, the table numbers has been added in the text as per valuable suggestion.

Reviewer question #4:  3.8 should not be separated from 3.7 in the Results section

  • Author response: Worthy reviewer, section 3.7 and 3.8 has been merged under section of “Biochemical parameters and biomarker study”

Reviewer question #5:  Figures 2 and 3, Lines 340 and 372 – fractions should be specified

  • Author response: Worthy reviewer, the fractions were specified accordingly.

Reviewer question #6:  Line 469 - Scopolamine in this study?

  • Author response: Worthy reviewer, the statement has been corrected accordingly as streptozotocin was used instead of scopolamine for induction of amnesia. Sorry for the mistake

Reviewer question #7:  Metformin should be added to 2.1. Chemicals

Author response: Worthy reviewer, metformin along with source has been added accordingly.

Reviewer question #8:  Lines 201,203,205,207,209, …….. – animals were not treated with hexane, chloroform, ethyl acetate, butanol,…….   ; they were treated with hexane, chloroform, ethyl acetate, butanol fraction

  • Author response: Worthy reviewer, the statement in the respective lines has been corrected accordingly.

Reviewer question #9:  Line 57, 63-64 – repetition

  • Author response: Worthy reviewer, the repetition was omitted accordingly.

Reviewer question #10:  Line 61 – diabetes mellitus (DM)

  • Author response: Worthy reviewer, it has been corrected accordingly.

Reviewer question #11:  Line 76 – Artemisia belongs to  Asteraceae

  • Author response: Worthy reviewer, the statement has been corrected accordingly.

Reviewer question #12:  Line 119 – ( instead of I

  • Author response: Worthy reviewer, the statement has been corrected accordingly as per valuable suggestion.

Reviewer question #13:  Line 251 – objects were identical or similar?

  • Author response: Worthy reviewer, the statement has been corrected accordingly as per valuable suggestion and the correct word id identical for this methodolgy.

Reviewer question #14:  Line 348 - in comparison to amnesic group 32.93%. – could be deleted

  • Author response: Worthy reviewer, deleted accordingly as per valuable suggestion.

Reviewer question #15:  Line 400 – FBG should be defined

  • Author response: Worthy reviewer, defined accordingly.

Reviewer question #16:  Lines 525-526 – abbreviations should be used

  • Author response: Worthy reviewer, abbreviation was used accordingly.

English

Reviewer question #1:  Line 70 – „The researchers are trying to isolate novel potent compounds with easy affordability and safety profile than synthetic drugs for low income patients [11].”

  • Author response: Worthy reviewer, the statement has been rephrased accordingly as per valuable suggestion.

Reviewer question #2:  Lines 209, 211, 359, 366, 402, 425, 432, 450 – change ethylacetae to ethyl acetate  

  • Author response: Worthy reviewer, corrected accordingly

Reviewer question #3:  Line 28 – streptozotocin-induced diabetic mice

  • Author response: Worthy reviewer, the statement has been corrected accordingly

Reviewer question #4:  Line 58 – epidemiological studies have revealed

  • Author response: Worthy reviewer, the statement has been corrected accordingly as per valuable suggestion

Reviewer question #5:  Line 63 - ACh possesses

  • Author response: Worthy reviewer, the statement has been corrected accordingly as per valuable suggestion

Reviewer question #6:  Table 4 Line 318 – “Crd-Am exhibited the major rise (PË‚0.01, PË‚0.001) in the spontaneous alternation at 100 and 200 mg/kg as compared to amnesic group that was found to be 57.13±1.59 (PË‚0.01, n=6) and 62.54±1.50 (PË‚0.001, n=6).” – what is meant by the major rise as some other extracts demonstrated higher effects?

Author response: Worthy reviewer, the statement has been corrected accordingly as per valuable suggestion and the statement of “major rise” has been changed with “an increase.”

Reviewer question #7:  Line 124 - AICl3 as quantifying

Author response: Worthy reviewer, the statement has been corrected accordingly as per valuable suggestion

Reviewer question #8:  Line 184 – in a citrate buffer

  • Author response: Worthy reviewer, the statement has been corrected accordingly as per valuable suggestion

Reviewer question #9:  Line 187 - Those animals having fasting blood glucose level higher than 250 mg/dL were subjected to assessment of memory [29].

Author response: Worthy reviewer, the statement has been corrected accordingly as per valuable suggestion

Reviewer question #10:  Table 2 – yield instead of yield

  • Author response: Worthy reviewer, corrected accordingly

Reviewer question #11:  Line 302 - respectivel against – please correct English

  • Author response: Worthy reviewer, the statement has been corrected accordingly

Reviewer question #12:  Line 487 – hippocampal

  • Author response: Worthy reviewer, corrected accordingly

Reviewer 3 Report

I have read the manuscript entitled “Effects of Artemisia macrocephala Jacquem on memory deficits and brain oxidative stress in streptoizocin induced diabetic mice”. The findings presented in this ms revealed that Artemisia macrocephala Jacquem enhances memory impairment associated with diabetes, elevated acetylcholine level, and decreased oxidative condition after chronic (4weeks) administration to mice. I think this work is interesting and the findings of this studies may enlarge the knowledge of the impact of Artemisia macrocephala on the cognitive dysfunction in STZ induced memory impairments. Despite the value of this study, there are some major and minor issues with the present work that need to be addressed, and these are my comments for the Authors’ consideration:

Major points:

  1. How many animals had proper STZ-induced diabetes and were subjected to assess memory impairments in the NOR and Y-maze tests? Authors do not show full statistical data which are obtained in Anova analysis of variance - this makes the results difficult to compare and
  2. Why did the Authors choose a different route of administration of “positive control” (intraperitoneally) and investigated Artemisia extracts (orally). Furthermore, there is no information on what exactly this 'positive control' is exactly – as I can only guess this may be groups treated with donepezil and Why these drugs were given i.p., while in clink they are given orally? What volume of test compound solutions were administered to mice - no information available in ms.
  3. In the figure 1 – NORT is marked only on 29th day of theexperiment, while Authors describe in the ms that NORT has been done twice to assess short and long-term memory pp6 ll248-249: “For short term memory, the test phase was performed 248 for 5 min after the sample administration while for long term memory; the test phase was 249 performed 24 h after the sample applications”. Moreover, in the results of NORT the phase T2 is skipped with T1- in the T1 phase the same objects are presented to animals, while in the T2 phase one of the objects is novel. There is a wrong symbol for “Familiar object A2” in the legend of Fig 2 (B) and Fig 3 (B).
  4. The results obtained for brain biomarkers should be described in a more precise way, the proper English should be used, except arrow symbols, that is, “↓ 2.98folds”. Moreover in this part of ms the Authors refer to the scopolamine data obtained in this study ( 16 ll 469 ) however this compound was not investigated by the Authors.
  5. Did Authors use the ARRIVE guidelines for reporting experiments using live animals as a checklist?
  6. There are numerous typos and language errors throughout the manuscript. A more thorough proof-reading is required.

Minor points

  1. The proper name of NORT is Novel Object Recognition test.
  2. The footnotes for Table 8 need to be corrected: There is no explanation for # symbol.7

Author Response

Reviewer 3:

I have read the manuscript entitled “Effects of Artemisia macrocephala Jacquem on memory deficits and brain oxidative stress in streptoizocin induced diabetic mice”. The findings presented in this ms revealed that Artemisia macrocephala Jacquem enhances memory impairment associated with diabetes, elevated acetylcholine level, and decreased oxidative condition after chronic (4weeks) administration to mice. I think this work is interesting and the findings of this studies may enlarge the knowledge of the impact of Artemisia macrocephala on the cognitive dysfunction in STZ induced memory impairments. Despite the value of this study, there are some major and minor issues with the present work that need to be addressed, and these are my comments for the Authors’ consideration:

Major points:

  1. Reviewer question #1: How many animals had proper STZ-induced diabetes and were subjected to assess memory impairments in the NOR and Y-maze tests? Authors do not show full statistical data which are obtained in Anova analysis of variance - this makes the results difficult to compare and

Author response: Worthy reviewer, the statement has been corrected accordingly as per valuable suggestion. A total of 96 animals has been used for the assessment of behavioral effects using Y-Maze and NORT. Worthy reviewer, the statistic has already been applied on the data you pointed out. The results are presented in Table and graph with statistical symbols.

  1. Reviewer question #2: Why did the Authors choose a different route of administration of “positive control” (intraperitoneally) and investigated Artemisia extracts (orally). Furthermore, there is no information on what exactly this 'positive control' is exactly – as I can only guess this may be groups treated with donepezil and Why these drugs were given i.p., while in clink they are given orally? What volume of test compound solutions were administered to mice - no information available in ms.

Author response: Worthy reviewer, the statement has been corrected accordingly as per valuable suggestion. The revised statement is now as follow

The animals in group A and B served as control and diabetic control group respectively. The animals in respective group C-N were administrated with the various doses via oral gavage (p.o.), whereas group O and P were positive control (donepezil and metformin) that received treatment via oral route.

  1. Reviewer question #3: In the figure 1 – NORT is marked only on 29th day of the experiment, while Authors describe in the ms that NORT has been done twice to assess short and long-term memory pp6 ll248-249: “For short term memory, the test phase was performed 248 for 5 min after the sample administration while for long term memory; the test phase was 249 performed 24 h after the sample applications”. Moreover, in the results of NORT the phase T2 is skipped with T1- in the T1 phase the same objects are presented to animals, while in the T2 phase one of the objects is novel. There is a wrong symbol for “Familiar object A2” in the legend of Fig 2 (B) and Fig 3 (B).

Author response: Worthy reviewer, the statement has been corrected accordingly as per valuable suggestion. On 29th day, short term memory was assessed while on 30th day long term memory was assessed. Figure 1, Figure 2 (B) and 3 (B) has been modified accordingly. Thanks for your positive input in correcting our mistakes.

  1. Reviewer question #4: The results obtained for brain biomarkers should be described in a more precise way, the proper English should be used, except arrow symbols, that is, “↓ 2.98folds”. Moreover in this part of ms the Authors refer to the scopolamine data obtained in this study ( 16 ll 469 ) however this compound was not investigated by the Authors.
  • Author response: Worthy reviewer, the whole section has been corrected accordingly with proper English as per valuable suggestion. Also thanks for pointing out the mistakes in the section.
  1. Reviewer question #5: Did Authors use the ARRIVE guidelines for reporting experiments using live animals as a checklist?
  • Author response: Worthy reviewer, the ARRIVE guidelines has been followed which were accordingly cited in the respective section as well.
  1. Reviewer question #6: There are numerous typos and language errors throughout the manuscript. A more thorough proof-reading is required.
  • Author response: Worthy reviewer, the manuscript has been revised taking help from language expert. Hopefully, now it will be ok.

Minor points

  1. Reviewer question #1: The proper name of NORT is Novel Object Recognition test.

Author response: Worthy reviewer, the statement has been corrected accordingly as per valuable suggestion

  1. Reviewer question #2: The footnotes for Table 8 need to be corrected: There is no explanation for # symbol.7

Author response: Worthy reviewer, on the suggestion of other two reviewer this table has now been moved to supplementary file as Table S4 where the required explanation has been provided accordingly as per valuable suggestion.

Round 2

Reviewer 2 Report

Line 123 – delete were

Line 151 – Similar fractions were combined and further subjected?

Line 270 - ; could be deleted

Line 278 – Biochemical

Figures 2C and 3C – pull down ###

Line 404 – At the same time, oxidative stress marker level….?

Line 469 – most could be deleted

Line 481 – evidenced

Line 506 – due to instead of despite

Line 515 – only abbreviations should be used

Lines 521-527 – exact values together with P and n should be omitted from the Discussion section